# Monitoring Pharmaceuticals and Personal Care Products in Drinking Water Samples by the LC-MS/MS Method to Estimate Their Potential Health Risk

**DOI:** 10.3390/molecules28155899

**Published:** 2023-08-05

**Authors:** Lucia Molnarova, Tatana Halesova, Marta Vaclavikova, Zuzana Bosakova

**Affiliations:** 1Department of Analytical Chemistry, Faculty of Science, Charles University, Hlavova 8, 128 43 Prague, Czech Republic; lucia.molnarova@natur.cuni.cz; 2ALS Czech Republic, Na Harfe 223/9, 190 00 Prague, Czech Republic; tatana.halesova@alsglobal.com (T.H.); marta.vaclavikova@alsglobal.com (M.V.)

**Keywords:** health risk, COVID-19 pandemic, drinking water, pharmaceuticals, personal care products, HPLC-MS/MS, direct injection

## Abstract

(1) The occurrence and accumulation of pharmaceuticals and personal care products in the environment are recognized scientific concerns. Many of these compounds are disposed of in an unchanged or metabolized form through sewage systems and wastewater treatment plants (WWTP). WWTP processes do not completely eliminate all active substances or their metabolites. Therefore, they systematically leach into the water system and are increasingly contaminating ground, surface, and drinking water, representing a health risk largely ignored by legislative bodies. Especially during the COVID-19 pandemic, a significantly larger amount of medicines and protective products were consumed. It is therefore likely that contamination of water sources has increased, and in the case of groundwater with a delayed effect. As a result, it is necessary to develop an accurate, rapid, and easily available method applicable to routine screening analyses of potable water to monitor and estimate their potential health risk. (2) A multi-residue UHPLC-MS/MS analytical method designed for the identification of 52 pharmaceutical products was developed and used to monitor their presence in drinking water. (3) The optimized method achieved good validation parameters, with recovery of 70–120% of most analytes and repeatability achieving results within 20%. In real samples of drinking water, at least one analyte above the limit of determination was detected in each of the 15 tap water and groundwater samples analyzed. (4) These findings highlight the need for legislation to address pharmaceutical contamination in the environment.

## 1. Introduction

Pharmaceuticals and personal care products (PPCPs) from various sources have become one of the major pollutants in the environment. Scientists and control laboratories are currently working on developing improved methods for detecting the presence of these drugs in the environment. Studies have shown that the main sources of this environmental contamination are hospitals, individuals’ use of over-the-counter drugs, and sewage treatment plants (WWTP). Unfortunately, existing WWTPs are not able to effectively remove and decompose drugs and their by-products due to insufficient treatment [1,2]. These pollutants and their metabolites found in drinking water can have harmful effects on both the endocrine system and reproductive development of humans. In addition, the potential for the development of antimicrobial resistance due to long-term exposure to these contaminants represents a significant public health concern [3,4].

A study from 2017 monitored the presence of the five most commonly used non-steroidal anti-inflammatory drugs in water sources at 29 locations in the Czech Republic. The results showed that ibuprofen was the most frequently found drug and in the highest concentrations, followed by naproxen, diclofenac, ketoprofen, and indomethacin [5]. Several other studies from Europe reported the detection of PPCPs in drinking and surface water samples [6]. Testing of drinking and bottled water in Portugal has revealed the presence of fluoxetine, ibuprofen, ketoprofen, and carbamazepine. Bottled water contained 30.6 ng L^−1^ of salicylic acid metabolites, while drinking water contained 66.0 ng L^−1^. Additionally, seawater samples contained paracetamol, ibuprofen, ketoprofen, metabolites of salicylic acid, fluoxetine, carbamazepine, and sulfamethoxazole [7]. The presence of pharmaceuticals and hormones in drinking, surface, and ground water in the metropolitan area of Turin, Italy has also been confirmed. Pharmaceuticals with the highest detected concentrations were atenolol, estrone, carbamazepine, ketoprofen, and diclofenac [8]. The surface water samples from six rivers and one lake in Slovenia were examined to determine the presence of various pharmaceuticals. Three common analytes were present in all samples: caffeine, irbesartan, and valsartan [9]. A study conducted in Paris, France monitored the occurrence of pharmaceuticals in the Seine river. The results showed relatively high concentrations of gabapentin, tramadol, oxazepam, diclofenac, and carbamazepine [10]. In the surface waters of Spain and Italy, the most frequently detected analgesic was ibuprofen, which was found in 17 out of 20 samples where concentrations reached up to 42 μg L^−1^. Lower concentrations of the caffeine metabolite paraxanthine, paracetamol, and carbamazepine were found in a smaller number of samples [11].

Effective detection and quantification methodologies are crucial to efficiently identify the persistence and ecotoxicity features of PPCPs in the environment [12]. The identification of (semi)volatile thermolabile compounds in the aqueous phase has seen extensive use of GC-MS techniques. In contrast, non-volatile, thermostable, and polar molecules may now be analyzed using LC-MS. Because of these obvious benefits, such as reduced sample pre-treatment and a broad ability to identify polar or thermally stable chemicals, LC-MS approaches have expanded the scope of GC-MS [13]. For the identification and measurement of these pollutants, triple quadrupole mass spectrometry in conjunction with liquid chromatography has proven to be an effective technique [14].

It is frequently essential to modify the sample and concentrate the analytes prior to the actual LC-MS detection due to the extremely low levels of the target analytes and the relatively large variety and complexity of the studied water matrices. The removal of solid particles from the matrix is usually the first step in sample preparation. This is achieved by filtering, utilizing filters consisting of glass microfibers (1 or 0.7 mm), nylon (0.45 mm), or cellulose (0.45 mm) [15]. Samples are frequently acidified or enriched with chelating agents (such as disodium salt of ethylenediaminetetraacetic acid, or EDTA) to avoid the breakdown of pharmaceuticals during transport and storage [7,16].

Water samples are filtered, extracted, and concentrated in various ways depending on the type of water matrix being tested (drinking, bottled, tap, utility, surface, subterranean, waste), as well as the sensitivity of the MS method. The direct injection (DI) technique is the only one that can be used effectively in some pure water analysis situations (drinking and bottled water). As a result, the water samples are not treated in any manner, only a combination of internal standards (ISTD) is added before the sample is filtered through a microfilter (such as a PTFE filter with a 0.2 μm porosity) and used for LC-MS analysis [17,18].

The quick, easy, cheap, effective, rugged, and safe (QuEChERS) extraction and purification method is another particular technique that is occasionally used in water analysis. The QuEChERS technique may be used to analyze polycyclic aromatic hydrocarbons, mycotoxins, pesticides, and pharmaceuticals in water and other complicated matrices [19,20].

Solid-phase extraction (SPE) is a method used worldwide to selectively isolate drugs from aqueous matrices. These plastic columns are filled with a particular sorbent. Polymer sorbents with reversed phases (such as Oasis HBL [21,22,23]) and ion-exchange characteristics are among the most widely utilized sorbents (e.g., Strata XC [7,9], and Oasis MCX [24]). SPE cleaning can be performed either manually with a vacuum manifold or automatically with an online link to the LC-MS system. By doing so, handling, labor, and time may be reduced while also improving the repeatability of the result [15].

PPCPs are a diverse category of molecules with a wide range of physical and chemical properties, as well as distinct structures and functional groups. High-performance liquid chromatography (HPLC) has been used more often in recent years, which has greatly increased separation efficiency and decreased the run time of separations [13].

There have been several multi-residue methods developed recently for the identification of tens to hundreds of PPCPs in various aqueous matrices, such as analytical LC–MS/MS screening techniques for 34 PPCPs in different environmental samples, including surface water samples, WWTP samples, and soil samples [25]. The screening of PPCPs in a conventional drinking water system was performed by using suspect and non-target analysis with liquid chromatography-tandem high-resolution mass spectrometry (LC-HRMS) and target analysis with LC-MS/MS [26]. An analytical method SPE-LC-MS/MS for the quantification of 15 EDCs (i.e., pharmaceuticals, hormones, plasticizers, and pesticides) revealed the occurrence of these contaminants in drinking water [27].

In this study, we describe the optimization of mobile phases and the shortened validation of a screening multi-residual LC-MS/MS method for the determination of 52 analytes of PPCPs in relatively clean water samples using direct injection (Appendix A). The PPCPs were taken into consideration for the current investigation, as they were positively identified in various studies focused on drinking water analyses and also due to their frequency of usage in the Czech Republic in previous years [28,29,30]. The file of targeted PPCPs includes the drugs in different therapeutical classes, such as antidiarrheal medication, anticoagulants, antihypertensives, antibiotics, chemotherapeutic agents, cytostatics, immunosuppressants, myorelaxants, non-steroidal antiphlogistic drugs, analgesics, psycholeptics, psychoanaleptics, antiasthmatics, and iodinated contrast media.

The goals of this method were to (a) minimize manual sample processing procedures to just add internal standards and (b) filter the sample before analysis to obtain the lowest detection and determination limits achievable. Limiting manual sample processing procedures in commercial laboratories reduces the risk of human errors and enables the implementation of automated sample preparation. Incorporation of appropriate internal standards into the analytical method helps correct variations in sample preparation and instrument response and ensures accurate and reliable quantitation. Additionally, a robust filtration process involving filtration of a 10 mL sample through a syringe with a cellulose filter directly into the vial helps remove particles and potential interferences from water samples. Optimizing the detection and determination limits of this analytical method involves using appropriate instrumentation and optimizing analytical parameters to achieve the highest sensitivity while maintaining acceptable precision and accuracy. In addition, the implementation of strict quality control measures ensures the accuracy and reliability of the analytical data. Therefore, method validation was performed to demonstrate that it meets the required standards for precision, accuracy, sensitivity, and selectivity. The assay is easily expandable and adaptable to a wider variety of analytes.

The second objective of this work was the application of the methodology to real samples collected in the Czech Republic during COVID-19 pandemic. The presence of selected PPCPs was investigated in treated tap water and potable groundwater samples.

## 2. Results and Discussion

### 2.1. Optimization of HPLC-MS/MS

For the optimization of the mass spectrometry parameters, a standard solution of each analyte (500 ng mL^−1^) with its corresponding internal standard was prepared in 50% methanol and MQ water, and directly injected into the MS/MS system.

By running a full scan mode chromatogram of single analytes, the precursor and product ions were selected for both identification and quantitation purposes (see Appendix A. The precursor ion was determined as a single protonated or deprotonated molecule mass. Next, the cone voltage and collision energy were optimized for each transition of individual analytes. A total of six (enalapril, flutamide, furosemide, gemfibrozil, hydrochlorothiazide, and chloramphenicol) out of 52 target analytes showed higher responses in negative electrospray ionization modes, the rest were analyzed in positive electrospray ionization mode. The operational parameters of the MS/MS detection are listed in Table 1.

Testing of different mobile phase compositions was performed by measuring the same fortified solvent blank spiked with the analyte standard at 0.1 ng L^−1^ in five replicates for each mobile phase. The seven most frequently used mobile phases (see Figure 1) were selected from the available literature to obtain high responses and symmetrical peaks of the studied analytes. [7,31,32]. The highest overall sum of peaks was obtained in the mobile phase composed of acetonitrile as organic modifier and 0.01% formic acid (Figure 1, column D). However, the peaks of 16 analytes (atenolol, buprenorphine, butorphanol, citalopram, cyclobenzaprine, enalapril, gabapentin, loperamide, metoprolol, mycophenolate mofetil, propranolol, sertraline, thebaine, tramadol, trimethoprim, and zolpidem) were distorted and exhibited tailing in this mobile phase composition. Replacing acetonitrile with methanol resulted in only a slight decrease in the total peak sum (Figure 1, column G) but led to a substantial improvement in the peak shapes of most problematic analytes (for an example, see Figure 2).

The inclusion of a very small amount of formic acid enhanced the retention, peak shape, and sensitivity of polar compounds, reducing tailing effects. On the other hand, methanol improved the elution of non-polar analytes, leading to increased solubility, improved peak widths, and symmetrical peaks. The mobile phase containing 0.01% formic acid and methanol provided the best results for all analytes. Introducing a gradient elution for analytes with very different physical and chemical properties significantly shortened the elution time and improved the shape of the peaks (for final separation conditions, see Table 1).

### 2.2. Validation of Method

The UHPLC-MS/MS method was validated for the determination of 52 substances (Appendix A) in drinking water. MQ water was used as the validation reference matrix. Both matrices (drinking water, MQ water) included in the validation were validated at two concentration levels (0.01 ng mL^−1^ and 0.1 ng mL^−1^), which were prepared in five replicates. The validation of the method was carried out with the aim of determining the basic validation characteristics, i.e., accuracy, precision, linearity, working range, limit of detection, limit of quantification, and selectivity. The target analytes were quantified by the external standard method, using isotopically labeled standards of drugs (ISTDs). ISTDs were added to both the fortified matrices and the analyzed samples at a level of 0.1 µg L^−1^.

Correction of analyte recovery and matrix effects was performed by conversion to the recovery of the assigned ISTD (Appendix A).

Since there is not a commercially available ISTD for every analyte included in the method, laboratory practice has shown that it is possible to substitute another ISTD for the exact corresponding isotopically labeled analog, based on the similarity of the molecular structure, the similarity of the retention time, and the similarity of the recoveries of the analytes and the ISTDs in individual matrices [33,34]. ISTDs were assigned to each target analyte based primarily on their corresponding behavior in the tested matrices.

#### 2.2.1. Selectivity

During the validation, no unwanted interferences were observed in the tested matrices which would lead to questioning the selectivity of the method (Appendix A).

#### 2.2.2. Limit of Detection (LOD) and Limit of Quantitation (LOQ)

When comparing the LOD and LOQ for the entire group of analytes (Table 2), relatively higher limits (> 0.01 ng mL^−1^) were obtained, for example, for the analyte caffeine, cyclobenzaprine, fluoxetine, iomeprole, and sertraline. Given that legally permissible hygienic limits are not (yet) set for medicinal products, the maximum possible permissible determination limits are not defined either. In addition to the substances listed above, low detection limits were obtained for the other substances, which are suitable for routine determination in a commercial laboratory environment.

#### 2.2.3. Linear and Working Range

For drinking samples, the linear and working range of the method is shown in Appendix A. The coefficients of determination (*R*^2^) are given as they were exported from the TargetLynx evaluation program in the given concentration range. If the concentration of analytes in water samples exceeds the level of 1 ng mL^−1^, the samples must be appropriately diluted and reanalyzed. Most of the tested analytes showed *R*^2^ > 0.999. The exception was the analyte caffeine, which, due to its lower sensitivity and higher limits of detection/quantification, also shows a narrower linear range compared to other analytes.

#### 2.2.4. Precision

The precision of the method, evaluated through the relative standard deviation RSD (%), showed that the RSD for most analytes, regardless of the tested matrix, did not exceed 20% (Appendix A). Higher RSD values were found for analytes that showed worse results for other validation parameters, e.g., higher LOD/LOQ. The higher RSD values, i.e., the greater dispersion of the determined results, were probably due to the fact that the spike level of 0.1 ng mL^−1^ was also very close to their limit of quantification for these problematic analytes, and the data obtained fluctuated more.

#### 2.2.5. Accuracy

Appendix A displays the accuracy of the method, expressed as recovery (%), and it can be observed from the data that the recovery for most analytes is between 70 and 120%. These values are similar to those in other studies of drinking water analyses [7,35]. A lower concentration level was prepared for analytes with a larger linear range. For some analytes (e.g., cyclobenzaprine, fluoxetine, gabapentin, iomeprole, capecitabine, caffeine, clofibric acid, loperamide, naproxen, paclitaxel, and sertraline), the recovery at the lower spike level is not reported because the LOQ of these analytes was higher than the concentration level tested. Higher or lower recovery values than recommended for the determination of individual analytes in fortified MQ water blanks and fortified drinking water samples may be caused by matrix effects due to solvent calibration evaluation.

### 2.3. Results of Drug Monitoring in Drinking Water Samples

The validated UHPLC-MS/MS method was used for monitoring drugs in drinking water collected in the territory of the Czech Republic, which was a combination of tap water and drinking groundwater. The results are graphically shown in Figure 3 and summarized in Table 3, and it can be seen that 12 of the 52 monitored drugs were found in water samples from the Czech Republic, where at least one analyte was determined in each sample.

The residual concentrations of targeted drugs ranged from below the detection limits to 204 ng L^−1^. Carbamazepine, gabapentin, and iodine substances were detected in drinking water samples above the level of 0.1 μg L^−1^, which is a value that is used as a limit and hygienically allowed in the determination of pesticides.

The analyte that was detected in practically all tested samples (14 positive out of 15 analyzed) across the Czech Republic was the iodine contrast agent, iomeprol. The average value of occurrence of this substance was 0.090 μg L^−1^. On the other hand, iopamidol was detected only in two groundwater samples (0.014 and 0.188 μg L^−1^). In the case of iodine substances, these are a group of pharmaceuticals used in hospitals in relatively high doses. When all types of water are contaminated with these substances, the processes applied at wastewater treatment plants, i.e., ozonation, UV radiation, chlorination, or microbial degradation, have no major effect on the removal of these substances [36]. The antiepileptic carbamazepine was detected as the second analyte in terms of the number of positively contaminated samples. It was determined in roughly half of the samples (7/15), at an average concentration of 0.122 μg L^−1^. In two samples of tap water, carbamazepine was detected even in several times higher concentrations (0.412 and 0.180 μg L^−1^) than in raw groundwater samples. In Portugal, China, and USA, drinking and bottled water have been both found to contain comparable or lower amounts of carbamazepine [7,35,37].

An investigation of the effectiveness of conventional drinking water treatment plants (DWTP) in the removal of various medications (including analgesics, NSAIDs, antibiotics, antiepileptics, beta-blockers, and lipid regulators), personal care products, and hormones was conducted in Spain. Ibuprofen and carbamazepine were present in treated water samples, but in small quantities (0.09–0.5 ng L^−1^) [38]. The frequent occurrence of carbamazepine and its metabolite is probably caused by poor removal efficiency in the water treatment process [35,39]. Gabapentin, another antiepileptic drug, was also detected in four samples with an average concentration value of 0.178 μg L^−1^. The antihypertensive agent valsartan (0.021 μg L^−1^) was detected in groundwater in relatively low concentration [39]. Furthermore, the pharmaceuticals in the samples determined in lower concentrations were gemfibrozil (0.091 μg L^−1^), tramadol (0.044 μg L^−1^), sulfamethoxazole (0.035 μg L^−1^) chloramphenicol (0.024 μg L^−1^), valsartan (0.021 μg L^−1^), sulfamethazine (0.011 μg L^−1^), and sotalol (0.011 μg L^−1^). Surprisingly, caffeine was detected in only one groundwater sample at a concentration of 0.034 μg L^−1^, which is probably due to the high limit of quantitation [40].

Additionally, further papers have revealed that several drugs had similar concentrations in drinking water to those found in our study, including caffeine, carbamazepine, and sulfamethoxazole [35,37,40]. Some studies monitoring the occurrence of drugs in drinking water, on the other hand, found none or in very low concentrations, below LOQ, of the monitored analytes detected in our study [29,41]. The occurrence of 17 pharmaceuticals in raw and treated drinking water was confirmed by a study conducted in Poland, where the drugs paracetamol (118.8 ng L^−1^) and ketoprofen (58.8 ng L^−1^) were found, in some cases even in higher concentrations than in raw drinking water. Diclofenac, flurbiprofen, ibuprofen, and naproxen were detected only in raw drinking water samples [30].

As is well known, drinking water is water that has undergone drinking water treatment and is suitable for ingestion by humans. Despite treatment, modest levels of pharmaceuticals and illicit drugs are still being found in drinking water at quantities in the ng L^−1^ range across Europe, America, and Asia [42,43]. The fact that such xenobiotics are not completely eliminated by existing drinking water treatment techniques raises concerns. In general, several techniques are used to turn raw water into drinkable water. To remove organic and inorganic impurities from water and sterilize for microorganisms, modern drinking water treatment techniques such as reverse osmosis, granular activated carbon (GAC), ultrafiltration, and chlorine gas are used [43,44]. The effectiveness of PPCPs and narcotic drug elimination depends on the water treatment techniques used. Advanced water treatments have been shown to be an effective approach to removing these micropollutants from drinking water; however, not all substance groups are affected by this fact, and the treatment method itself may have an impact on the amount of narcotic drugs present in the water [39,42,43].

## 3. Materials and Methods

### 3.1. Chemicals and Reagents

A list of analytical standards and isotopically labelled internal standards are included in Appendix A. LC-MS grade methanol and acetonitrile were obtained from Honeywell (Honeywell, Charlotte, NC, USA). Formic and acetic acids were MS grade and purchased from Sigma-Aldrich^®^ (Sigma-Aldrich^®^, Darmstadt, Germany). Ultrapure water (MQ) was supplied through a Milli-Q water system (Millipore^®^, Billerica, MA, USA).

### 3.2. Sample Collection

The sampling points were chosen to cover different populated areas. Tap water samples were collected in 12 cities in the Czech Republic with a population of more than 50,000, including two samples in different locations in the capital Prague (TW1 and TW2). Two groundwater samples were sampled from clean drinking springs.

Tap water and groundwater samples (1 L) from 15 individual sampling points were taken in polypropylene (PP) bottles that had been pre-cleaned with deionized water, during April and May 2020. Tap water was allowed to run to waste for about two minutes before collection was started. The bottles were rinsed with deionized water prior to use. The collected samples were immediately stored in a freezer.

### 3.3. Stock Solutions, Calibration Standards, and Quality Control (QC) Samples

Individual stock solutions were prepared in methanol or acetonitrile (10 μg mL^−1^, 100 μg mL^−1^, and 1000 μg mL^−1^). A mixed stock solution of analytes and ISTDs was diluted from these stock solutions at a concentration of 10 ng mL^−1^. All solutions were stored at 4 °C. Calibration standards were freshly prepared by spiking MQ and blank drinking water with mixed stock solution of analytes and ISTDs to obtain final concentrations of 0.15, 0.0025, 0.0050, 0.010, 0.025, 0.050, 0.075, 0.10, 0.25, 0.50, and 1.00 ng mL^−1^.

### 3.4. Chromatographic Conditions

The analyses were performed using an UPLC I-class liquid chromatograph (Waters, Milford, MA, USA) coupled to an MS analyzer XEVO TQ-S (Waters, USA). The operational parameters of the LC-MS/MS method are listed in Table 1. Chromatographic separations were performed using an Acquity UPLC BEH C18 Chromatography Column (1.7 μm, 2.1 × 100 mm) with an Acquity UPLC BEH C18 (1.7 μm) pre-column (Waters, USA), heated at 40 °C. Mobile phase A consisted of formic acid (0.01%, *v*/*v*) in Milli-Q water, and mobile phase B was 100% methanol. The acquisition was performed using electrospray ionization, in both positive and negative mode. The analyses were performed in multiple reaction monitoring (MRM) mode. Two MRM transitions, quantification and qualification, were selected for each analyte. The MRM transitions of each compound, their cone and respective collision energies, as well as their retention time are presented in Appendix A.

### 3.5. Preparation of Blank Samples

A 10 mL water sample was measured into a plastic centrifuge tube with a calibrated automatic pipette. ISTD solution with a concentration of 10 ng mL^−1^ was then added to the water samples to create two concentration levels, 0.01 ng mL^−1^ and 0.1 ng mL^−1^. Samples of the tested matrices were prepared in triplicate for each concentration level. The sample was then shaken by hand and filtered through 0.20 µm disk cellulose filters. During sample filtration, the first 6 mL of the sample was filtered into the waste, and the subsequent aliquot portion of the sample (approx. 1 mL) was filtered into a dark 2 mL vial intended for LC-MS analysis.

### 3.6. Preparation of Laboratory Control Samples (LCS)

A 10 mL quantity of MQ water was measured into a plastic centrifuge tube with a calibrated automatic pipette. In order to perform the analysis correctly, an amount of standard stock solution (spike) with a concentration of 10 ng mL^−1^ was first added to the measured volume of MQ water. The sample was properly shaken, and only then was a spike of the ISTD stock solution with a concentration of 10 ng mL^−1^ added. LCS samples were prepared at two concentration levels of 0.01 ng mL^−1^ and 0.1 ng mL^−1^ and were further filtered through 0.20 µm disk cellulose filters. During sample filtration, the first 6 mL of the sample was filtered into the waste, and the subsequent aliquot portion of the sample (approx. 1 mL) was filtered into a dark 2 mL vial intended for LC-MS analysis.

### 3.7. Preparation of Fortified Matrices for Validation

The samples were prepared using real water samples according to the procedure mentioned above. As with LCS sample preparation, ISTD and analyte stock solutions were added to the water samples at certain concentration levels. As part of the validation, spiking of water samples (10 mL) was performed at two concentration levels, 0.01 ng mL^−1^ and 0.1 ng mL^−1^, by removing the appropriate volume from the stock solution of standards. Samples of the test matrices were prepared in five replicates for each concentration level.

### 3.8. Method Validation

Using optimized chromatographic conditions, the established HPLC approach has been validated in terms of selectivity, limit of detection and limit of quantification, linear and working range, precision, and accuracy.

#### 3.8.1. Selectivity

Verification of the selectivity of the LC-MS method was performed by monitoring 2 MRM transitions of each analyte using MS detection at a certain retention time of the given analyte. The retention time and ratio of MRM transitions must be maintained in a defined ratio for individual analytes in the standard and in the sample, and was monitored by evaluation software (MassLynx 4.2, Waters, Milford, MA, USA) as part of the integration of quantification and confirmation peaks of individual target analytes.

#### 3.8.2. Limit of Detection and Limit of Quantification

The limit of detection (LOD) and limit of quantification (LOQ) values were determined simultaneously with verification of the correctness and accuracy of the method by analyzing five fortified MQ water matrix blanks and five fortified drinking water samples at two concentration levels (0.01 ng mL^−1^ and 0.1 ng mL^−1^).

#### 3.8.3. Linear and Working Range

The linearity and thus the working range of the method was tested using mixed solutions of drug standards at ten concentration levels in the range of 0.0025–1.00 ng mL^−1^. In the indicated concentration range, most analytes showed a non-linear dependence, and therefore a second-order polynomial dependence was chosen for all analytes and evaluation.

#### 3.8.4. Precision

The precision of the method is expressed using the repeatability of the determination of individual analytes and is calculated as the relative standard deviation of the determination (RSD) under repeatability conditions. The repeatability of the method was determined based on LC-MS measurements of five fortified blanks of MQ water and drinking water at two concentration levels (0.01 ng mL^−1^ and 0.1 ng mL^−1^). Repeatability is expressed as the RSD value [%] for each analyte in Equation (1):(1)$$RSD\;[\%]=\;SD/\bar{x},$$
where SD is the sample standard deviation and $\bar{x}$ is the average of the measured concentrations of the given analyte at one concentration level.

#### 3.8.5. Accuracy

The verification of the accuracy of the method was carried out at the same time as the repeatability testing and is expressed using the recovery calculated as the ratio of the detected concentration of the analyte in the fortified sample to the accepted reference value (amount of added analyte) in Equation (2):(2)Recovery% Rec=cicspike
where *c*_i_ is the measured analyte content and *c*_spike_ is the concentration with addition of analytes (sample enrichment level).

The recovery of individual analytes was performed on five fortified matrix blanks of MQ water and five fortified drinking water samples at two concentration levels (0.01 ng mL^−1^ and 0.1 ng mL^−1^). The yield of individual analytes was converted to the yield of the assigned ISTD.

## 4. Conclusions

The work was focused on the optimization and validation of the multi-residue analytical UHPLC-MS/MS method, which served for the determination of 52 drugs. As part of the method optimization, various modifiers of the mobile phase were tested, to improve the analysis of problematic analytes. The peaks of 16 analytes were distorted and showed tailing. A mobile phase containing 0.01% formic acid and organic solvent methanol provided the best results and improved the peak shapes.

The monitoring of 52 drug analytes in real samples of drinking water in the Czech Republic was carried out. Fifteen samples of tap water and drinking groundwater were analyzed, where at least one analyte above the detection limit was determined in each sample. The analyte detected in practically all samples was iomeprol, i.e., an iodinated contrast agent with an average concentration value of 0.093 μg L^−1^. Carbamazepine was found in seven samples with an average concentration of 0.182 μg L^−1^. Gabapentin was present in four samples with the highest average concentration value of 0.178 μg L^−1^. Furthermore, iopamidol (0.101 μg L^−1^), gemfibrozil (0.091 μg L^−1^), tramadol (0.036 μg L^−1^), sulfamethoxazole (0.035 μg L^−1^), chloramphenicol (0.024 μg L^−1^), valsartan (0.021 μg L^−1^), sulfamethazine (0.011 μg L^−1^), and sotalol (0.011 μg L^−1^) were determined in the samples at relatively low concentration levels.

According to the obtained results, pharmaceuticals represent the same burden on the environment as, for example, pesticides, plasticizers, and hormones. These biologically active compounds are present in water ecosystem and adversely affect aquatic organisms and the environment. While the impact varies depending on the specific compounds and environmental conditions, both pharmaceuticals and pesticides pose a risk to aquatic life, disrupting food chains, and affecting ecosystem health [27,45,46]. In the future, continuous attention should also be paid to these contaminants, legislative limits should be set, but above all, processes should be tested and defined that will lead to the degradation and gradual removal of these substances from water and the environment. The quality of ground and surface water is also important from a socioeconomic point of view. Ensuring clean and safe water sources is essential for public health, agriculture, and environmental sustainability. Taking proactive measures to address water contamination and degradation is crucial for promoting economic growth, protecting public health, and maintaining social balance.

## Figures and Tables

**Figure 1 molecules-28-05899-f001:**
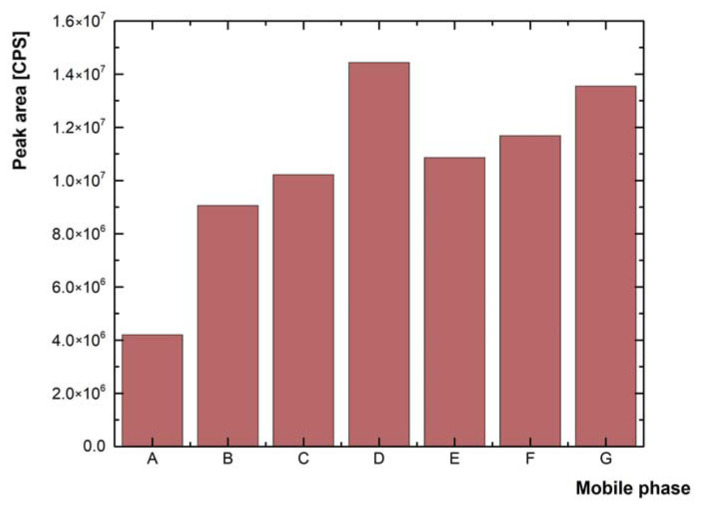
Comparison of the sum of the peak areas of all analytes on mobile phase composition. (Peak area is expressed in units of Counts per Second (CPS)). Mobile phases: A—acetic acid 0.1%/acetonitrile, B—formic acid 0.1%/acetonitrile, C—formic acid 0.05%/acetonitrile, D—formic acid 0.01%/acetonitrile, E—formic acid 0.1%/methanol, F—formic acid 0.05%/methanol, G—formic acid 0.01%/methanol.

**Figure 2 molecules-28-05899-f002:**
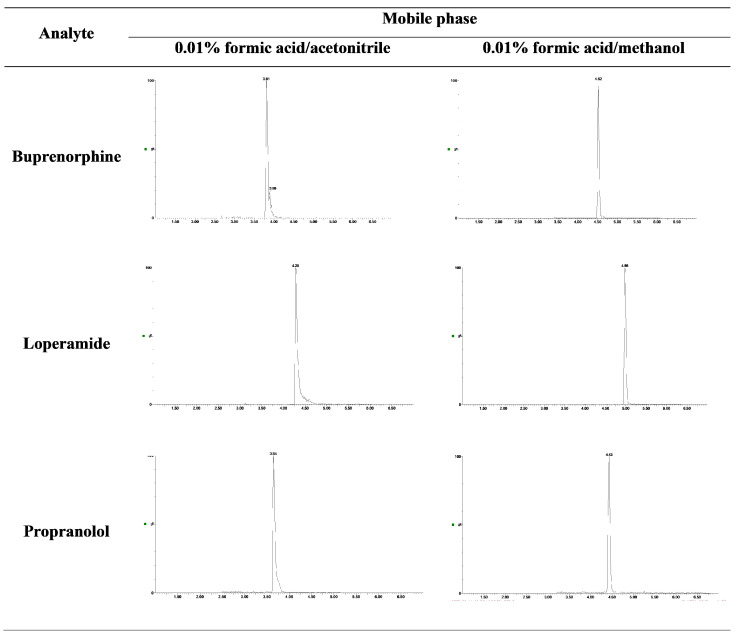
The effect of the type of organic modifier on the peak shape of the analytes.

**Figure 3 molecules-28-05899-f003:**
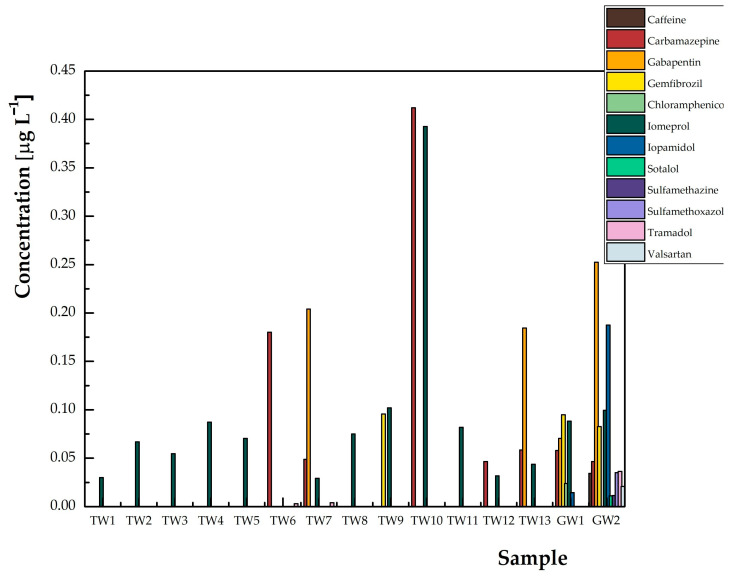
Overview of the occurrence of drug analytes determined in tap water (TW) and drinking groundwater (GW) samples in the Czech Republic.

**Table 1 molecules-28-05899-t001:** Final operating parameters of LC-MS/MS method.

**MS conditions**	Heat block temperature	150 °C							
	Column voltage	0.75 kV							
	Desolvation temperature	600 °C							
	Desolvation gas flow	800 L hod^−1^							
	Cone gas flow	150 L hod^−1^							
	Nebulizer gas	7.0 bar							
	Collision gas flow	0.19 L min^−1^							
	MRM window	0.4 min							
**LC conditions**	Column		Acquity UPLC BEH C18 (2.1 mm × 100 mm × 1.7 µm)						
	Column temperature		40 °C						
	Injection volume		50 µL						
	Flow rate		0.4 mL min^−1^						
	Total run time		9 min						
	Mobile phase	A	0.01% formic acid in Milli-Q water						
		B	methanol						
	Gradient profile	Time (min)	0	0.5	5.0	5.1	7.0	7.1	9.0
		A (%)	98	98	5	0	0	98	98
		B (%)	2	2	95	100	100	2	2

**Table 2 molecules-28-05899-t002:** Overview of values of limits of detection and determination for individual analytes in MQ and drinking water.

Analyte	Average [ng L^−1^]	SD [ng L^−1^]	LOD [ng L^−1^]	LOQ [ng L^−1^]
Anastrozole	9.57	0.26	0.96	2.89
Atenolol	11.47	0.88	3.30	9.89
Azathioprine	9.87	0.48	1.80	5.39
Bezafibrate	8.88	0.86	3.22	9.66
Buprenorphine	12.04	0.40	1.51	4.52
Butorphanol	11.92	1.12	4.21	12.62
Caffeine	90.94	15.60	58.45	175.36
Capecitabine	88.04	17.02	63.78	191.35
Carbamazepine	9.54	0.36	1.36	4.07
Citalopram	13.39	1.37	5.14	15.41
Clofibric acid	66.05	12.59	47.19	141.55
Cyclobenzaprine	67.31	7.70	28.85	86.56
Cyclophosphamide	9.26	0.56	2.10	6.30
Diazepam	9.79	0.19	0.72	2.16
Diclofenac	8.12	2.27	8.50	25.49
Enalapril	9.89	0.46	1.71	5.14
Fluoxetine	93.57	17.08	64.00	192.01
Flutamide	8.69	0.55	2.05	6.16
Furosemide	8.14	1.47	5.49	16.48
Gabapentin	88.59	2.97	11.14	33.41
Gemfibrozil	10.59	1.05	3.92	11.75
Hydrochlorothiazide	8.44	1.38	5.18	15.54
Chloramfenicol	11.42	1.39	5.19	15.58
Ifosfamide	10.02	0.62	2.31	6.93
Indomethacin	8.44	0.70	2.62	7.85
Iomeprol	8.13	1.09	4.10	12.30
Iopamidol	9.96	0.92	3.46	10.38
Iopromide	157.95	25.87	96.93	290.78
Ketoprofen	7.79	0.58	2.18	6.55
Lincomycin	10.28	0.64	2.41	7.22
Loperamide	145.68	19.76	74.04	222.11
Metoprolol	11.55	0.90	3.36	10.09
Metronidazole	9.90	0.19	0.70	2.09
Mycophenolate Mofetil	8.10	1.50	5.62	16.86
Naproxen	79.33	6.37	23.87	71.60
Oxazepam	9.56	0.32	1.18	3.55
Paclitaxel	64.46	12.44	46.60	139.81
Paracetamol	9.25	1.20	4.49	13.48
Piroxicam	9.93	0.31	1.15	3.45
Propranolol	9.42	0.63	2.35	7.04
Salbutamol	11.96	0.94	3.50	10.51
Sertraline	73.65	14.35	53.78	161.34
Sotalol	10.97	0.81	3.03	9.09
Sulfamethazine	9.68	0.24	0.90	2.71
Sulfamethoxazole	9.46	0.55	2.05	6.14
Terbutaline	11.93	1.37	5.13	15.38
Thebaine	11.92	1.38	5.18	15.55
Tramadol	11.54	1.09	4.08	12.24
Trimethoprim	8.04	1.19	4.47	13.40
Valsartan	9.09	1.27	4.77	14.31
Warfarin	8.04	0.51	1.92	5.75
Zolpidem	9.29	0.68	2.55	7.64

**Table 3 molecules-28-05899-t003:** Overview of concentrations of determined drug analytes in tap water (TW) and drinking groundwater (GW) in the Czech Republic.

Analyte	Concentration in Samples [μg L^−1^]														
	TW1	TW2	TW3	TW4	TW5	TW6	TW7	TW8	TW9	TW10	TW11	TW12	TW13	GW1	GW2
Caffeine	n. d.	n. d.	n. d.	n. d.	n. d.	n. d.	n. d.	n. d.	n. d.	n. d.	n. d.	n. d.	n. d.	n. d.	<LOQ
Carbamazepine	0.180	0.049	n. d.	n. d.	n. d.	n. d.	n. d.	n. d.	n. d.	0.412	n. d.	0.047	0.058	0.058	0.047
Gabapentin	n. d.	0.204	n. d.	n. d.	n. d.	n. d.	n. d.	n. d.	n. d.	n. d.	n. d.	n. d.	0.185	0.071	0.252
Gemfibrozil	n. d.	n. d.	n. d.	n. d.	n. d.	n. d.	n. d.	n. d.	0.096	n. d.	n. d.	n. d.	n. d.	0.095	0.083
Chloramphenicol	n. d.	n. d.	n. d.	n. d.	n. d.	n. d.	n. d.	n. d.	n. d.	n. d.	n. d.	n. d.	n. d.	0.024	n. d.
Iomeprol	n. d.	0.029	0.082	0.067	0.055	0.087	0.070	0.075	0.102	0.393	0.082	0.032	0.044	0.088	0.099
Iopamidol	n. d.	n. d.	n. d.	n. d.	n. d.	n. d.	n. d.	n. d.	n. d.	n. d.	n. d.	n. d.	n. d.	0.014	0.188
Sotalol	n. d.	n. d.	n. d.	n. d.	n. d.	n. d.	n. d.	n. d.	n. d.	n. d.	n. d.	n. d.	n. d.	n. d.	0.011
Sulfamethazine	n. d.	n. d.	n. d.	n. d.	n. d.	n. d.	n. d.	n. d.	n. d.	n. d.	n. d.	n. d.	n. d.	n. d.	0.011
Sulfamethoxazole	n. d.	n. d.	n. d.	n. d.	n. d.	n. d.	n. d.	n. d.	n. d.	n. d.	n. d.	n. d.	n. d.	n. d.	0.035
Tramadol	<LOQ	<LOQ	n. d.	n. d.	n. d.	n. d.	n. d.	n. d.	n. d.	n. d.	n. d.	n. d.	n. d.	n. d.	0.036
Valsartan	n. d.	n. d.	n. d.	n. d.	n. d.	n. d.	n. d.	n. d.	n. d.	n. d.	n. d.	n. d.	n. d.	n. d.	0.021

n.d.—not detected.

## Data Availability

Not applicable.

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
