# Peer review of "Monitoring Pharmaceuticals and Personal Care Products in Drinking Water Samples by the LC-MS/MS Method to Estimate Their Potential Health Risk"

_molecules, 2023, doi:10.3390/molecules28155899_

Round 1
Reviewer 1 Report
This paper presents a multi-residue UHPLC–MS/MS analytical method designed to identify 52 pharmaceutical products and its application in monitoring their presence in drinking water samples. While the optimized method demonstrated good validation parameters, further research is required to address the legislative gaps and effectively mitigate pharmaceutical contamination in the environment.
As a peer reviewer, I find this paper to be a valuable contribution to the field of environmental monitoring and risk assessment of pharmaceuticals and personal care products in drinking water. The abstract provides a clear and concise overview of the research objectives, methodology, and key findings. The significance of the study is well established, highlighting the potential health risks associated with the contamination of water sources. The development of a multi-residue UHPLC–MS/MS method to identify pharmaceutical products in drinking water samples is commendable, and the validation parameters indicate the reliability and robustness of the analytical approach.
However, I recommend that the authors address several points to improve the clarity and depth of the paper. First, it would be beneficial to provide more details about the experimental setup, including sample collection procedures, sample preparation techniques, and chromatographic conditions. This information would enhance the reproducibility and comparability of the study, enabling other researchers to replicate the method.
This paper presents a well-executed study that successfully developed a multi-residue UHPLC–MS/MS method for the identification of pharmaceuticals in drinking water samples. The findings highlight the urgent need for legislative measures to address pharmaceutical contamination in the environment. With the incorporation of the suggested revisions, this paper will significantly enhance our understanding of the potential health risks associated with pharmaceuticals and personal care products in drinking water and contribute to the ongoing efforts to safeguard water quality.
As a peer reviewer, I recommend accepting the paper despite significant revisions being necessary. This implies that the research holds potential and contributes to the field, but several important changes are needed to enhance its quality. Accepting the paper with major revisions allows the authors to address the reviewer's comments and improve the paper's overall clarity, methodology, and impact before final publication. Therefore, I recommend accepting this paper with major revisions.
- Various studies detecting PPCPs in water sources are mentioned, it would be beneficial to provide the publication details (authors, title, journal, and year) for each study.
- Line 68: The sentence "The identification of (semi)volatile thermostable compounds in the aqueous phase has seen extensive use of GC-MS techniques" seems to contain a typographical error, as "thermostable" might be meant to be "thermolabile" to indicate compounds that are easily degraded by heat. Please clarify and correct this.
- Line 126: The sentence "The goals of this method were to a) minimize manual sample processing procedures to just add internal standards and b) filtering the sample before analysis to obtain the lowest detection and determination limits achievable" could be improved by providing more specific information on how the goals were achieved. For example, mentioning the specific steps taken to minimize manual processing or describing the filtration process in more detail would provide clarity and enhance the understanding of the methodology.
- In the manuscript, there are several instances where the text references an error message, "Error! Reference source not found." It appears that these errors are due to missing or incorrect citations.
- In the second paragraph of the Conclusions section, it would be helpful to provide more specific details about the improvements achieved through the optimization of the method. How did the modifiers of the mobile phase enhance the analysis of problematic analytes? Please provide specific examples or results to support this statement.
- In the third paragraph, when discussing the presence of specific analytes in the samples, it would be beneficial to mention the units of measurement for the concentration values (e.g., μg/L or ng/mL Line 404 and other’too) for clarity and consistency. Additionally, please ensure that the average concentration values are correctly stated for each analyte.
- The statement "Based on the obtained results, pharmaceuticals pose the same burden to the environment as, for example, pesticides" could benefit from further elaboration. It would be helpful to explain the basis for this comparison and provide additional evidence or references to support this claim.
- In the last sentence of the Conclusions section, when mentioning the importance of water quality from a socioeconomic perspective, it would be valuable to provide some specific examples or reasons to support this statement.
Overall, the manuscript presents a comprehensive study on the optimization and validation of a multi-residue analytical method for the detection of 52 drugs in drinking water samples. The authors have made valuable contributions to the field by addressing the need for improved methodologies to monitor pharmaceutical contaminants in the environment. The findings highlight the presence of several drugs in the tested samples, with iomeprol, carbamazepine and gabapentin being the most frequently detected analytes.
In order to improve the manuscript's clarity, accuracy, and overall impact, it is essential to incorporate the suggested major revisions. By addressing the identified areas for improvement, the manuscript will undergo significant enhancements that will positively contribute to its quality and appeal to readers. Taking these major revisions into consideration will ultimately result in a more polished and compelling manuscript that is well-positioned for final acceptance.
Reviewer 2 Report
Few dozen years ago the presence of some pharmaceutical residuals in ground and drinking water was detected first time. Since that time this fact seems to be rather important in numerous aspects so that it can be neglected. Hence, the manuscript considered maybe classified as significant contribution to our knowledge about this problem.
Specific features of the manuscript submitted has are the following: At first, it is the distribution of information between the Main Text and the Supplement. A lot of important information is presented in the Supplement only: it is the list of pharmaceuticals considered, the list of standards, the set of MS/MS parameters, the set of LOD and LOQ values for all the analytes, as well the ranges of linearity, and characteristics of repeatability. Some of this information (e.g., lists of compounds/standards and the sets of LOD/LOQ values) seem to be presented not in the Supplement, but in the Main Text. Without this data the information being included in the Main Text looks (sorry) like simplified wordy description of the serious problem. However, the reviewer leaves this question to the Editor to decide.
Some of pharmaceuticals selected for consideration in this manuscript exist as hydrates. For example, caffeine can be used in the analytical practice both as anhydrous substance, and monohydrate. I propose to the Author to check all the compounds listed in Table 1 (Supplement) and indicate those of them which were used in hydrated forms. Please do not neglect this recommendation.
Some small corrections can be recommended, as well:
- Page 2, line 82: EDTA as chelating agent means not the acid, but disodium salt of ethylenediaminetetraacetic acid. It seems to be better to correct this chemical name.
- Page 2, lines 92-94. The first sentence of the last paragraph contains the decoding of QuEChERS abbreviations. If so, it seems not necessary to explain the same in the next sentence; it may be deleted.
- Page 3, line 100: Small correction: restore the spacer between XC and references [7, 9].
- Line 273: Correct misprinting: should be “carbamazepine”.
- Section 3, subsection 3.4 includes the paraphrase of chromatographic conditions presented in Table 1. It may be strongly shortened and/or united with this Table.
- Lines 406 – 408, relations (3.1) and (3.2): By definition, a percent is a hundredth of a number. If so, not necessary to indicate the factor “x100” in the right parts of these relations. The reviewer understands that indicating of this factor is a very common practice, but the correct representation of the formula is more important.
The corrections mentions should be classified as “minor (or moderate) revision”. After that the manuscript can be recommended for publication in “Molecules”.
No comments
